# Audio Prototypical Network for Controllable Music Recommendation

## Abstract

Traditional recommendation systems represent user preferences in dense representations obtained through black-box encoder models. While these models often provide strong recommendation performance, they lack interpretability for users, leaving users unable to understand or control the system's modeling of their preferences. This limitation is especially challenging in music recommendation, where user preferences are highly personal and often evolve based on nuanced qualities like mood, genre, tempo, or instrumentation. In this paper, we propose an audio prototypical network for controllable music recommendation. This network expresses user preferences in terms of prototypes representative of semantically meaningful features pertaining to musical qualities. We show that the model obtains competitive recommendation performance compared to popular baseline models while also providing interpretable and controllable user profiles.

## 1 Introduction

Modern recommender systems often rely on techniques such as collaborative filtering methods, which represent users with opaque numerical embeddings that are difficult for their users to interpret and not meant to control recommendations. While previous work aims to improve the scrutability of such systems by using keyword tags or natural language summaries to describe user preferences, such approaches are not universally applicable across domains. For instance, Siebrasse & Wald-Fuhrmann (2023) demonstrate that using broad genres to describe someone's musical taste can be misleading, as users with similar genre profiles may still have vastly different preferences. Their study shows that sub-genres, more closely tied to specific artists and musical elements, provide a more accurate representation of individual taste.

In light of these challenges, our work focuses on capturing user preferences through listenable audio clips, which transparently reflect the system's inferred understanding of their musical tastes. This encoding makes the system's assumptions more interpretable and empowers users by allowing them to fully scrutinize and correct their profiles, offering control over how their preferences are represented and over their proposed recommendations.

We introduce APRON: Audio PROtotypical Network for music recommendation, where prototypes are listenable audio clips. We showcase the difference between a traditional recommendation system and APRON in Figure 1. APRON draws inspiration from prototypical networks (e.g. ProtoPNET) (Chen et al., 2019; Donnelly et al., 2022; Willard et al., 2024; Zinemanas et al., 2021; Alonso-Jiménez et al., 2024; Heinrich et al., 2024), which are widely used in the Explainable AI (XAI) literature.

APRON leverages an attention mechanism to create a weighted combination of prototype representations of users' historical interactions, ensuring an interpretable user representation. Furthermore, by constraining the inferred prototype distribution to that of the recommended songs, we enable a fully steerable system, allowing users to scrutinize and adjust their profiles through simple modification of prototype weights.

We demonstrate that our proposed methodology significantly enhances the controllability of the system's recommendations while maintaining performance comparable to fully black-box models. To evaluate controllability, we simulate user updates to their profiles, such as adding or removing

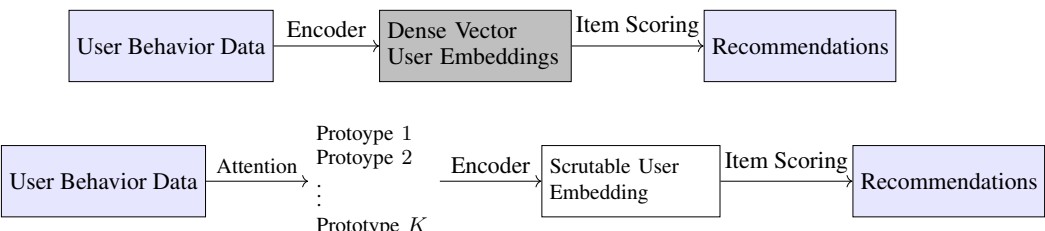

Figure 1: **(Top)** Classical Recommendation System Pipeline. **(Bottom)** The high-level pipeline for APRON. The User behavior Data (the audio features) are expressed in terms of listenable prototypes to create a scrutable user embedding. This embedding is then converted into recommendations.

prototypes, and measure the differences in recommendations between the original and modified profiles. Additionally, we conduct various ablation studies to validate our design choices.

We summarize our contributions as follows.

- We propose APRON, a prototypical network for music that expresses the overall user preferences using prototypes composed of listenable audio clips.
- We show that our model achieves a good controllability-accuracy tradeoff on the Million Song Dataset (MSD) (Bertin-Mahieux et al., 2011). APRON obtains similar performance as several strong baselines while enabling users to control their recommendations directly by altering their learnt representation.
- Since APRON relies on (song) features to obtain user profiles, we show that it can produce good-quality recommendations even when users listen to *cold-start* (unseen) songs.

## 2 RELATED WORK ON SCRUTABLE RECOMMENDER SYSTEMS

Explainable recommendation has become an increasingly important topic as recommender systems grow more complex and opaque. Traditionally, explainability has been approached post-hoc, using explanations based on models' features (Ning & Karypis, 2011; Shimizu et al., 2022; Vijayaraghavan & Mohapatra, 2024; Wang et al., 2018; Zhang et al., 2020) or more recently on LLM-produced explanations (Lubos et al., 2024; Luo et al., 2024). However, as noted earlier, these explanations might not be *actionable* by users or contain *truthful* information (Huang et al., 2023). On the other hand, scrutable recommender systems present the model's inferred user profile in a human-understandable and editable manner, enabling user interventions to directly influence the system's recommendations. This enhances actionability and truthfulness by allowing users to make meaningful changes that are transparently reflected in the system's behavior. Although holding many desirable properties, such systems have primarily been explored through the use of keywords or tags, which allow users to personalize their experience by selecting from a predefined collection (Green et al., 2009; Lubos et al., 2024; Moses & Babu, 2016; Balog et al., 2019). Unfortunately, representing a user's taste profile in this way can be limiting, as users may have to parse through an excessive collection of tags if they want to effectively customize their experience. More recently, scrutable systems have shifted towards using natural language summaries to represent users, offering an alternative to keyword-based personalization (Radlinski et al., 2022; Ramos et al., 2024). Instead of relying on keywords/tags, these systems generate a personalized summary using natural text. While this approach works well for domains suited to textual descriptions—such as movies, TV shows, or restaurants—it may not translate as effectively to other domains, like music or fashion, where user preferences might be difficult to express easily through text and could be better expressed through other mediums, such as audio or images. This highlights the need for more flexible approaches that can adapt scrutability to wider content types.

In this work, we address both limitations by enabling prototypes to attend to items in the user's history, allowing us to maintain scrutability while offering a more personalized experience. Additionally, to the best of our knowledge, this is the first work that allows users to scrutinize their recommendations using song-based prototypes, offering a much more suitable medium for music recommendations.

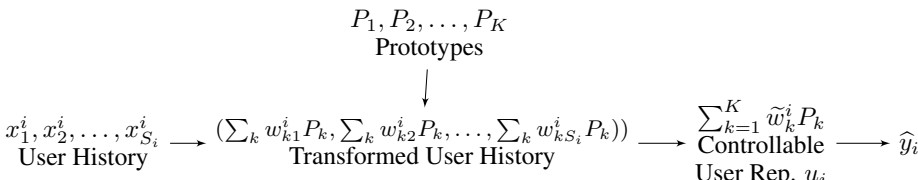

Figure 2: The pipeline of our model APRON. First of all the user history is transformed through calculating the attention weights $w_{kj}^i$ through the prototypes $P_1, \ldots, P_K$. Then the user representation $u_i$ is obtained by summing over the songs. Then finally the model output $\widehat{y}_i$ is calculated through transforming this user representation $u_i$ through the feedforward network $f(.)$.

## 3 METHODOLOGY: PROTOTYPE-BASED CONTROLLABLE USER REPRESENTATION

Our main goal is to express a user's historical interactions in terms of listenable prototypes, each associated with distinct musical concepts. In our experiments, musical concepts are encoded with tags corresponding to musical qualities (e.g. era, instrumentation, mood). Let us denote the user history for $i$'th user as,

$$\mathcal{X}_i = \{x_1^i, x_2^i, \ldots, x_{S_i}^i\}, \tag{1}$$

where $S_i$ and $x_j^i \in \mathbb{R}^D$ respectively denote the total number of songs listened by user $i$, and the D-dimensional encoding of the $j$'th song listened by the $i$'th user. A reasonable way to construct the profile $u_i$ for user $i$ is by summing the representations of songs the user has listened to in the past,

$$u_i = \sum_{j=1}^{S_i} x_j^i. \tag{2}$$

Such representations could then be processed by an encoder which directly provides recommendations. However, to impose a controllability constraint on the user profile, we constrain each song representation in terms of prototypes $\{P_1, \ldots, P_K\}$, such that:

$$x_j^i = \sum_{k=1}^{K} w_{kj}^i P_k, \tag{3}$$

where $P_k \in \mathbb{R}^D$ is the prototype that corresponds to the $k$'th musical tag. Each tag corresponds to a musical concept (e.g. rock, jazz, 90s, country, instrumental – more generally tags correspond to musical qualities). Note that each song can have more than a single tag (e.g. an instrumental song with two associated genres such as country and ballad). The weights $w_{kj}^i$ are parametrized using an attention layer,

$$w_{kj}^i = \frac{\exp\left((P_k A_k^p)^\top (x_j^i A_j^x)\right)}{\sum_{k'}^{K} \exp\left((P_{k'} A_{k'}^p)^\top (x_j^i A_j^x)\right)}, \tag{4}$$

where $A_k^p \in \mathbb{R}^{D \times D'}$, $A_j^x \in \mathbb{R}^{D \times D'}$ are learnable parameter matrices. Each user profile is then modelled as

$$u_i = \sum_{k=1}^{K} \underbrace{\sum_{j=1}^{S_i} \frac{\exp\left((P_k A_k^p)^\top (x_j^i A_j^x)\right)}{\sum_{k'}^{K} \exp\left((P_{k'} A_{k'}^p)^\top (x_j^i A_j^x)\right)}}_{:=\widetilde{w}_k^i} (P_k A_k^p) = \sum_{k=1}^{K} \widetilde{w}_k^i \widetilde{P}_k, \tag{5}$$

Note that unlike the Eq. 3 the prototypes $P_k$ are transformed as well, such that we use the result of the vector-matrix product $P_k A_k^p$ as the Value vector in the attention calculation (similar to the standard query-key-value attention formulation). One difference from the standard query-key-value attention formulation is that we use the same learnable matrix for the key and value, since we observed that it results in a more controllable model.

# 4 TRAINING OBJECTIVES

After describing the process adopted in APRON to create the user profile, we now discuss how to generate predictions for training. The output distribution over song recommendations $y_i \in \mathcal{S}^L$ (where $L$ is the number of songs in the catalog, and $\mathcal{S}^L$ denotes an L dimensional probability simplex) is computed by obtaining the interpretable user profile from Eq. 3, through a series of feed-forward layers denoted with $f(.)$ followed by a softmax activation as:

$$\widehat{y}_i = a(f(u_i)), \tag{6}$$

where $a(.)$ is an activation function such as the Softmax or the Sigmoid. We describe the overall pipeline in Figure 2.

For training the full system, we now demonstrate the three objectives employed as below.

**Recommendation Objective.** We train this system with a recommendation system loss that aims to minimize the divergence

$$\mathcal{L}_{\text{RecSys}} = d(y \parallel \widehat{y}_i). \tag{7}$$

The divergence is typically chosen as negative binary cross-entropy loss.

**Controllability Objective.** In addition to the recommendation system loss, to allow the system to be controllable, we construct a loss objective that minimizes the divergence between the aggregate prototype weights $\widetilde{w}_k^i$ and tag distribution that corresponds to the model output. We express this controllability loss $\mathcal{L}_{\text{controllability}}$ as follows:

$$\mathcal{L}_{\text{controllability}} = d(\widetilde{w}^i \parallel T(\widehat{y}_i)), \tag{8}$$

where $T(\cdot)$ is a counting function that obtains the tag distribution given the songs selected with $\widehat{y}_i$. This loss imposes the constraint that, for user $i$, the tag distribution $T(\widehat{y}_i)$ that corresponds to the recommendation output of the model $\widehat{y}_i$ is as close as possible to the user's distribution over tag prototypes $\widetilde{w}^i \in \mathbb{R}^K$. For the choice of the divergence metric, we emprically observe that the Hellinger distance gives the best performance. Therefore in our experiments this is what we use for the controllability loss:

$$\mathcal{L}_{\text{controllability}} = \sum_{i=1}^{N} \frac{1}{\sqrt{2}} \sqrt{\sum_{k=1}^{K} \left( \sqrt{\widetilde{w}_k^i} - \sqrt{T(\widehat{y}_i)_k} \right)^2}, \tag{9}$$

**Prototype-separability Objective.** We include a prototype-separability loss to make the prototypes as representative and distinct as possible of the associated music tags. For this, we enforce the transformed prototypes $\widetilde{P}_k$ to be classified as the associated tag, after passing these vectors through a linear layer $\phi(\cdot) : D' \to K$. The corresponding loss is as follows:

$$\mathcal{L}_{\text{prototype-sep}} = d(\phi(\widehat{P}_k) \parallel e_k), \tag{10}$$

where $e_k$ is the unit-vector that corresponds to the $k$'th tag, and for $d(.\parallel.)$ we used the standard cross-entropy loss for multi-way classification. We observed that this loss helps in avoiding solutions where the transformed prototypes collapse to very similar vectors.

Finally, the overall training objective $\mathcal{L}$ is defined as a weighted sum of the above three objectives as follows, with relative strengths $\lambda_1, \lambda_2$.

$$\mathcal{L} = \mathcal{L}_{\text{RecSys}} + \lambda_1 \mathcal{L}_{\text{prototype-sep}} + \lambda_2 \mathcal{L}_{\text{controllability}}. \tag{11}$$

# 5 EXPERIMENTS

In this section, we evaluate the recommendation system performance of APRON along with other baseline models applicable for music recommendation. We also provide experimental results for controllability analysis of APRON.

## 5.1 EXPERIMENTAL SETUP

**Dataset and Evaluation Protocol.** We conduct our experiments with the MSD and follow the same data preprocessing procedure as in Liang et al. (2018) which only keeps the users who at least listened to 20 songs and the songs that are listened to by at least 200 users. Before this filtering stage we also removed the songs from the dataset for which we do not have the audio files. Our dataset consists of 40,940 songs, 469,432 train users, 50,000 validation users and 50,000 test users. We conduct our evaluation in terms of strong generalization in which training, validation and test sets have disjoint users. We report the Normalized Discounted Cumulative Gain (*NDCG@100*) as well as Recall (*Recall@20*, *Recall@50*)as they are the standard performance metrics in recommendation literature.

**Tags and Prototype Generation.** We select the prototypes to correspond to the 88 most commonly used song-level tags according to the Last.fm Dataset (Bertin-Mahieux et al., 2011). Tags fall into 4 major groups: era, genre, mood, and instrumentation. To create listenable prototypes, we generate 30-second long prototype songs using the MusicGen-Large-3.3B model (Copet et al., 2023) by querying *"[TAG] song"* (if the tag describes a quality of the song, e.g. 90s song, rock song) or *"a song played by [TAG]."* (if the tag describes instrumentation). Generated prototypes can be found in supplementary material.

**Music Feature Extractor.** We extract music features for each song in the dataset and prototype songs with the MERT-v1-330M model (LI et al., 2024). For a song of an arbitrary length, MERT-v1-330M model outputs an encoding in the shape of *(Time (T), Layers (N), Dimension (D))*. We take the average over T axis to reduce the dimensionality to *(N, D)*, which are *(25, 1024)* for the MERT-v1-330M model. We use the last representation layer in our experiments, which observed to give the best performance in terms of representation performance. In the end we have 1024 dimensional feature representations for each song in the dataset and prototype songs.

**Baselines.** As baselines, we use MultiDAE, MultiVAE (Liang et al., 2018), RecVAE (Shenbin et al., 2020) and MacridVAE, SEM-MacridVAE (Wang et al., 2023) with our data split. We could not directly use the numbers from the corresponding papers as the version of the dataset does not contain the audio files for 200 songs audio files, and we have therefore run the baselines ourselves. For the MultiVAE, MultiDAE, MacridVAE and SEM-MacridVAE we use SEM-MacridVAE's official codebase and for the RecVAE we use RecVAE's official codebase.

**Implementation Details.** In our experiments, when implementing the attention mechanism to express each song in terms of prototypes in Eq. 4, we use multihead-attention. This results in the following way of calculating the protoype weights for each song:

$$w_{kj,h}^i = \frac{\exp\left((P_{k,h}A_{k,h}^p)^\top (x_j^i A_{j,h}^x)\right)}{\sum_{k'}^K \exp\left((P_{k',h}A_{k',h}^p)^\top (x_j^i A_{j,h}^x)\right)}, \tag{12}$$

where we learn a matrix $A_{k,h}^p \in \mathbb{R}^{(D/nh)\times D'}$, $A_j^x \in \mathbb{R}^{(D/nh)\times D'}$, for each head $h$. The user profile is then calculated as,

$$u_{i,h} = \sum_{k=1}^K \sum_{j=1}^{S_i} \underbrace{\frac{\exp\left((P_{k,h}A_{k,h}^p)^\top (x_j^i A_{j,h}^x)\right)}{\sum_{k'}^K \exp\left((P_{k',h}A_{k',h}^p)^\top (x_j^i A_{j,h}^x)\right)}}_{\widehat{w}_{k,h}^i} (P_{k,h}A_{k,h}^p) = \sum_{k=1}^K w_{k,h}^i \widetilde{P}_{k,h}. \tag{13}$$

Note that $P_{k,h}$, is obtained by dividing the prototype vector into $H$ equal-length chunks. Then to obtain the final user profile $u_i$, we concatenate over the head dimension $h$, such that,

$$u_i = \text{Concatenate}([u_{i,1}, u_{i,2}, \ldots, u_{i,H}]), \tag{14}$$

where $H$ is the number of attention heads. We observe that increasing the number of parallel attention heads improves the recommendation performance while slightly decreasing the controllability metrics that we define in the next subsection. We explore the trade-off in Section 5.3.

The code for our implementation of APRON is available on our anonymous repository[1].

---

[1] https://anonymous.4open.science/r/apron_iclr2025-A2D3

Table 1: Comparison of recommendation performance of APRON and baselines.

| Method | Recall@20 | Recall@50 | NDCG@100 |
|---|---|---|---|
| MultiDAE | 0.253 | 0.355 | 0.300 |
| MultiVAE | 0.264 | 0.366 | 0.315 |
| RecVAE | 0.275 | 0.373 | 0.325 |
| MacridVAE | 0.276 | 0.369 | 0.330 |
| SEM-MacridVAE | 0.290 | 0.384 | 0.343 |
| APRON (Ours) | 0.273 | 0.371 | 0.321 |

**Controllability Metrics.** Besides the recommendation system performance, we also define a controllability metric based on NDCG as defined follows. For a specific tag $\tau$, we define the tag-wise $\mathrm{DCG}_t @k$ as follows (the subscript $t$ is used to denote tag-wise DCG),

$$\mathrm{DCG}_t @k(\tau) = \sum_{i=1}^{k} \frac{\mathbf{I}(\tau \in T(y_i))}{\log_2(i+1)}, \tag{15}$$

where $T(.)$ extracts the tag information that corresponds to song song $y_i$, and $\mathbf{I}(.)$ denotes the indicator function. That is, if the tag $\tau$ is contained in the tags of the song $y_i$ (denoted with $T(y_i)$, the indicator function $\mathbf{I}(.)$ returns 1).

Then we calculate NDCG for all users in $U_\tau$, where $U_\tau$ denotes the set of users having items with tag $\tau$:

$$\mathrm{NDCG}_t @k(\tau) = \frac{1}{|U_\tau|} \sum_{u \in U_\tau} \sum_{i=1}^{k} \frac{\mathbf{I}(\tau \in T(i))}{\log_2(i+1)} \tag{16}$$

We define the controllability metric ($\Delta @k(\tau)$) to measure the interpretability performance of our system. We calculate the change ($\Delta @k$) between the full (using all of the templates, denoted with a superscript $F$) and modified (we denote with a superscript $M$).

$$\Delta @k(\tau) = \mathrm{NDCG}_t^F @k(\tau) - \mathrm{NDCG}_t^M @k(\tau) \tag{17}$$

When we drop attention weights, we allow using all of the prototypes except the prototype that corresponds to the tag $\tau$. When we increase the attention weights, for each user if the attention score is smaller than a certain threshold $th$, we increase it to $th$. (We do this experiment for $th = 0.5, 0.7$)

In this calculation we filter the users with having 0 in both term since every user does not contribute to each tag. Results, averaged over all tags, are presented in Table 2. We furthermore provide an analysis to breakdown the contribution of each tag in Figures 3, 4.

## 5.2 RECOMMENDATION PERFORMANCE

In Table 1, we compare the recommendation performance of APRON and several baselines introduced in the previous section. We evaluate recommendation performance under strong-generalization (i.e for users not seen during training). We observe that APRON with an attention mechanism with 16 parallel head ($H = 16$) is able to obtain competitive results in terms of NDCG. We note that APRON is outperformed only by the SEM-MacridVAE baseline. This is encouraging since it implies that the proposed modelling, while constrained to represent users using music prototypes and the combination of losses, leads to strong recommendation performance. Of course, the goal of APRON is to provide scrutable encodings, which is a property other baselines fail to effectively capture (see Section 5.3)

## 5.3 ASSESSING THE CONTROLLABILITY OF APRON

To assess the controllability of APRON, we conduct an experiment where we manipulate the attention weights $\widehat{w}_k^i$ for $k \in \{1, \ldots, K\}$ that correspond to the different musical tags. The expectation

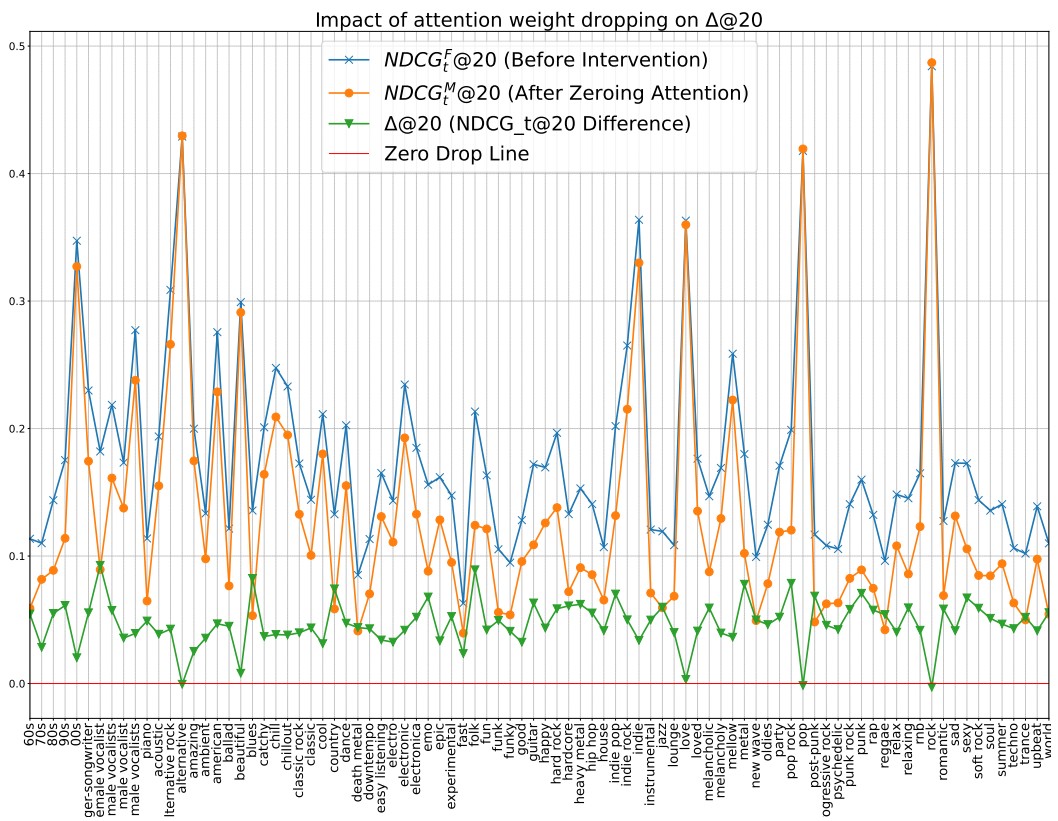

Figure 3: Tag based controllability of APRON (for attention weight reduction): We observe that for almost all tags reducing the associated attention weight $\widehat{w}_k^i$ results in reduction in $NDCG_t$. This experiment is conducted for $H = 16$.

is that if, for instance, the weight corresponding to tag $k$ is lowered, songs associated with this tag would be less likely to be recommended. We showcase this in Figure 3 where we systematically lower the weight associated with a tag and evaluate its effect on the recommendation quality. We observe that for almost all tags, reducing the attention weight to zero $\widetilde{w}_k^i$ for the tag $k$ results in a drop in $NDCG_t$ for that particular tag (using the metric defined in Eq. 17). We observe that tags that yield a large $NDCG_t$ value are more likely to still appear at the model output even if the attention weights are dropped. These correspond to generic tags such as 'alternative,' 'love,' or 'pop.' These songs are likely to have other tags associated with them. Therefore, we hypothesize that reducing attention weights corresponding to these tags does not decrease the recommendation of these songs, as they likely have other prototypes promoting them.

Additionally, we have conducted an experiment where we have increased attention weights of tags. For this, for each tag, for every user, we have increased the weights $\widetilde{w}_k^i$ for tag $k$ to 0.5 if we observe that the weight is lower than 0.5, and did not modify it if the weight is already larger than 0.5. In Figure 4 we showcase the results, and observe that APRON can increase model output for most tags. We see that especially for similar tags (e.g. female vocalist-female vocalists, alternative rock-alternative, funk-fun, relax-relaxing) we see this intervention might result in the opposite effect. But this is probably due to the confounding effect caused in this case by the prototype for a similar tag. In future work, this can be alleviated by using prototype-generated songs that are highly distinct from each other.

Furthermore, in Table 2, we explore the tradeoff between performance and controllability when the number of parallel attention heads is changed, and also compare the level of controllability achieved with APRON with that of SEM-MacridVAE. In Table 3, we report the average increase $NDCG_t$ for the case $H = 16$.

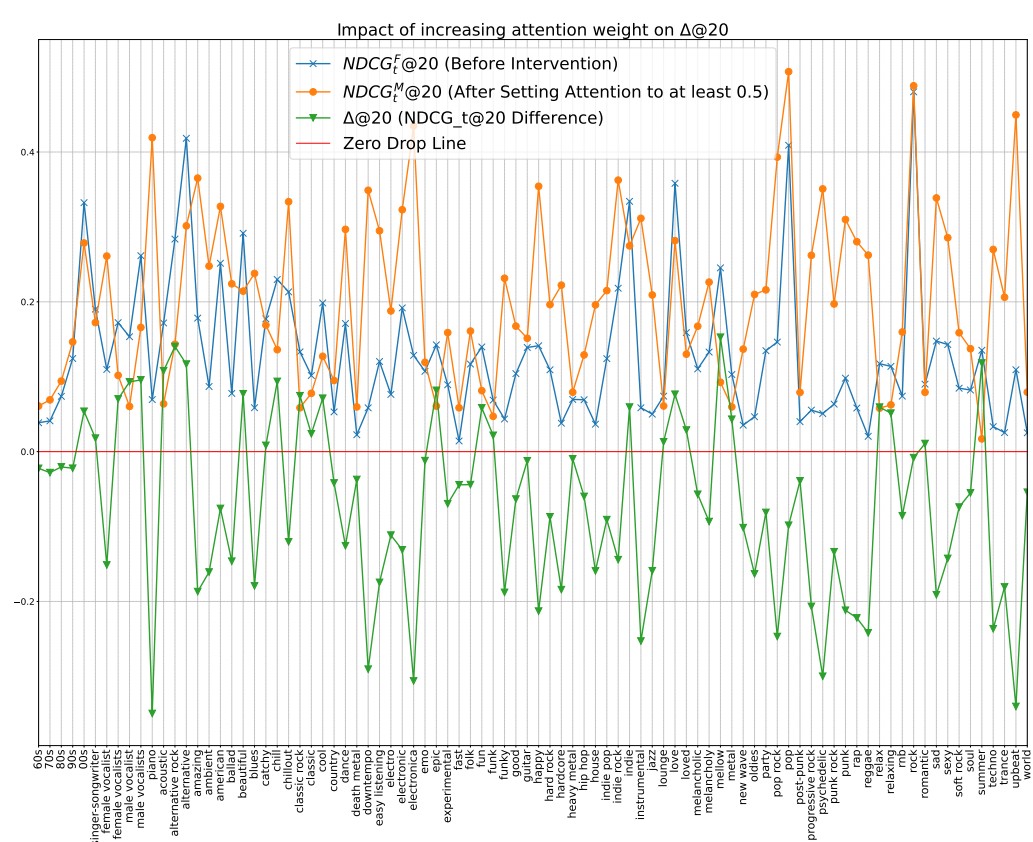

Figure 4: Tag based controllability of APRON (for attention weight increase): We observe that for most tags increasing the associated attention weight $\widehat{w}_k^i$ results in increase in $NDCG_t$. This experiment is conducted for $H = 16$, and the attention weights for the corresponding tag is set to 0.5.

Table 2: The performance trade-off between the model performance (NDCG) and the level of controllability ($\Delta$@20) when the number of heads $H$ is changed, for the experiment where we reduce the attention weights.

| Method | $H$ (# Heads) | $\Delta$@20 | % $\Delta$@20 | Recall@20 | Recall@50 | NDCG@100 |
|---|---|---|---|---|---|---|
| SEM-Macrid VAE | N/A | $-0.000\,54$ | $-0.30$ | 0.290 | 0.384 | 0.343 |
| APRON | 16 | $0.046\,87$ | 31.60 | 0.273 | 0.371 | 0.321 |
| APRON | 8 | $0.056\,03$ | 37.22 | 0.268 | 0.366 | 0.316 |
| APRON | 1 | $0.057\,02$ | 37.99 | 0.228 | 0.315 | 0.274 |

Table 3: The changes observed in $NDCG_t$ when the attention weights are increased.

| Method | $th$ (Threshold) | $H$ (# Heads) | $\Delta$@20 | % $\Delta$@20 |
|---|---|---|---|---|
| APRON | 0.5 | 16 | $-0.071$ | $-132.80$ |
| APRON | 0.7 | 16 | $-0.073$ | $-144.11$ |

## 5.4 ABLATION OF TRAINING LOSS COMPONENTS

To better understand the effects of the loss components $\mathcal{L}_{\text{RecSys}}$, $\mathcal{L}_{\text{rototype-sep}}$, and $\mathcal{L}_{\text{controllability}}$, we retrained APRON by changing the values of the tradeoff parameters associated with $\mathcal{L}_{\text{prototype-sep}}$ ($\lambda_1$)

and $\mathcal{L}_{\text{controllability}}$ ($\lambda_2$). We present these results in Table 4, where we compare the recommendation system performance and the controllability of the model for different $\lambda_1$, and $\lambda_2$ values. We observe that when the system is trained only with the Recommendation loss ($\lambda_1 = 0$, $\lambda_2 = 0$) both the recommendation system performance and the controllability are lower. We observe that the biggest gain in performance is obtained with the controllability loss (the case $\lambda_1 = 0$, $\lambda_2 = 0.005$). However, with the addition of the prototype-separability loss (the case $\lambda_1 = 1$, $\lambda_2 = 0.005$) we observe that the recommendation performance and the controllability metric is further improved. We also observe that these two losses also act as a regularizer improving the recommendation performance as well as the controllability metrics.

Finally, we would like to note that we have tried also the case, $\lambda_1 = 1$, and $\lambda_2 = 0$, but this did not result in stable training, as the transformed prototypes $\widetilde{P}_k$ all converged to non-separable vectors. We have also observed inferior performance for the case $\lambda_1 = 0.1$, $\lambda_2 = 0$, compared to the last row of Table 4.

Table 4: Ablation study with respect to the controllability-loss (the strength of which measured by $\lambda_1$), and the prototype separability loss (the strength of which measured by $\lambda_2$).

| Method | $\lambda_1$ | $\lambda_2$ | Recall@20 | Recall@50 | NDCG@100 | $\Delta$@20 |
|--------|------|------|-----------|-----------|----------|--------|
| APRON | 0 | 0 | 0.246 | 0.335 | 0.293 | 0.000 10 |
| APRON | 0.1 | 0 | 0.245 | 0.334 | 0.292 | $-0.001\,00$ |
| APRON | 0 | 0.005 | 0.266 | 0.362 | 0.315 | 0.036 46 |
| APRON | 1 | 0.005 | 0.273 | 0.371 | 0.321 | 0.046 87 |

## 5.5 Encoding Unseen Songs

As we have mentioned in the introduction, a significant advantage that comes with APRON compared to the other baselines is that, because APRON works with song features directly, it is able to encode songs that were not previously seen during training (called *cold-start items* in the recommendation systems).

To evaluate the performance in the case where we encode unseen songs, we create an additional control set that consists of 16,575 users who have listened to 9,313 additional songs besides the 40,940 that were used to train APRON. We made sure that each of these new songs were listened to at least 150 times. We have then created recommendations for these 16,575 users from the original training set that consists of 40,940 songs.

As other baselines work with a fixed item-set (that is, they require additional training to encode the new songs), we were not able to provide any baseline for this new evaluation set.

Results are presented in Table 5. As seen from the results, although we encode 9,313 new songs that the system had not been exposed to earlier, we observe that in terms of Recall, our model is able to remain competitive with the numbers reported in Table 1 for the baseline methods such as MultiDAE, MultiVAE, RecVAE and MacridVAE. We note that this experiment does not explicitly test for creating recommendations for unseen songs. However, we show here that the model is able to encode new songs. With a modification to the output loss function, such that at the output we calculate a similarity function (e.g. cosine similarity) between song embeddings and the user embeddings, the model can be adapted to produce recommendations for new songs also.

This therefore showcases the potential for APRON to be used for zero-shot generalization to new songs, for providing recommendations on unseen songs without needing to retrain the system. Moreover, we observe that APRON is still able to be similarly controllable when compared to the case where the testing is carried out on already seen songs.

## 6 Limitations and discussion on impact

The music datasets with user interaction information are hard to obtain. For the audio features corresponding to the Million Song Dataset we had to resort to peer sharing as many other researchers

Table 5: The recommendation performance and controllability metrics when we encode unseen songs with APRON.

| Method | Dataset | $H$ | $\Delta@20$ | % $\Delta@20$ | Recall@20 | Recall@50 | NDCG@100 |
|--------|---------|-----|-------------|---------------|-----------|-----------|----------|
| APRON  | Unseen  | 16  | 0.03931     | 27.31         | 0.282     | 0.372     | 0.258    |
| APRON  | Unseen  | 8   | 0.04594     | 31.83         | 0.276     | 0.364     | 0.252    |
| APRON  | Unseen  | 1   | 0.05136     | 35.22         | 0.222     | 0.298     | 0.206    |

in the Music Information Retrieval Domain (Kim et al., 2023). For this reason, we were not able to evaluate APRON on other datasets. We have also only investigated working with fixed prototypes, and did not experiment with modifying the subtle musical qualities in order to model the user preferences. The framework we propose in this paper however supports this, and in future works, we would like to explore editing the audio for prototypes directly, and study the behavior of the model with user studies. We will also work on generating and selecting higher quality prototypes.

We would like to state that this work aims to render the recommendation systems more transparent and controllable. Therefore we do not believe that our work has an obvious negative social impact. On the contrary, our methodology can be used to understand the biases in recommendation systems and to mitigate them.

## 7 CONCLUSIONS

In this paper, we have proposed APRON, a prototypical network for music recommendations. Our experiments on the Million Song Dataset show that APRON can produce controllable recommendations (more controllable compared to SEM-Macrid VAE, for example) while maintaining competitive recommendation performance with other baselines. Moreover, APRON can encode unseen songs; it is able to incorporate information from users who have listened to songs that had not been listened to before and to produce recommendations. All in all, APRON is a new form of scrutable recommendation system which directly exposes the user taste, paving the way for domain specific scrutable models that captures the feature level information. As future work, we would like to also apply APRON on other application domains where prototypes can be used to encode fundamental notions pertaining to items (e.g. clothing recommendation).

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
