# OpenReview forum: "Audio Prototypical Network for Controllable Music Recommendation"
_ICLR.cc/2025/Conference — Submitted to ICLR 2025_

### Official Review · Reviewer_PDqh · 2024-10-16

**Soundness:** 3
**Presentation:** 2
**Contribution:** 2
**Rating:** 3
**Confidence:** 4

**Summary:**

The paper proposes a "prototypical network" for music recommendation, which basically expresses user preferences in terms of a combination of listenable prototypes (i.e., short musical snippets). This is argued to be a more interpretable and controllable form of music recommendation as the tags (and snippets based on them) are interpretable and weighted preferences are controllable by the user.

**Strengths:**

The idea of using listenable prototypes to improve the interpretability of music recommendation is (to me) original, at least in the context of music recommendation.

**Weaknesses:**

Ultimately the prototypes did not strike me as convincing, and I didn't see additional value in them compared to more traditional tags. The generated audio did not seem to be faithful to the tags (e.g. music with male vocalist didn't have any vocalist -- this was just the first one I tried). So I'm not really convinced that "listenability" really adds to the usefulness of the method compared to a (presumably more trivial) approach purely based on tags. I'm not sure I generally see the value of a generative approach in this context, compared to e.g. extractively choosing snippets from songs a user likes. As a user it would be difficult for me to express my preferences in terms of these listenable snippets which often did not sound like "real" music.

Performance is also probably a sticking point. The authors acknowledge that the method is competitive with, but not stronger than baseline methods, and offer increased controllability to overcome this shortcoming. But I think that requires that the controllability / interpretability claims are totally convincing, which they aren't quite yet.

**Questions:**

-- Why does the approach need to be generative (abstractive) rather than extractive?

-- Is it possible to more directly compare against the strongest baselines on MSD? I understand there were some issues here since not all the songs contain audio and the dataset had to be sampled (and baselines rerun) but I am maybe missing a summary of whether the performance is really close to SotA for this dataset.

-- From the user perspective, what is the value of this system compared to just surfacing the tags themselves? To the extent that the tags *don't* match the snippets, the system isn't faithful and will confuse users. To the extent that the tags *do* match the snippets, the snippets are (arguably) redundant.

---

> ### Author Response · Authors · 2024-11-28
>
> We would like to thank the reviewer for their time creating their constructive feedback. Below are our responses to individual comments:
>
> Weakness 1 & Question 1 (Prototype selection process):  In our method, we used tag-based prototypes to create user profiles in terms of tags to control the user profiles. A user would be able to listen to the prototypes that correspond to the learnt profile and remove/add prototypes to steer the system in the direction of their own preferences. We would like to note that we used a generative model because we have the tag information in our dataset and via a generative model we can generate listenable prototypes that correspond to a particular tag. This way we associate the information with listenable prototypes.
>
> Weakness 2 (Performance): We are using fixed audio embeddings for items to remain controllable through the optimization and this restricts the performance. We currently do not see a comparable method that obtains similar levels of controllability in the current setup.
>
> Question 2 (Data splits): Other methods are also using the same procedure to create the splits which are keeping the songs at least listened 200 times and users who have listened to at least 20 songs. We followed the exact same procedure as the baseline results are very similar to the original results. We only had to remove 200 songs which we don’t have access to raw audio files. We observe for instance that EASE obtains better recommendation performance, however it is not a controllable model.
>
> Question 3 (Prototypes and tags):  As asked we have replied to the reviewer above, the purpose behind using prototypes is that we would like to be able to capture various musical nuances that can only be represented with the audio modality. (E.g. certain rhythmic, melodic, harmonic patterns. We agree with the reviewer that in the current version, since we only have one template per tag, the current system is not significantly different from tag-based prototypes. However, we believe it served as a proof-of-concept for future improvements that use multiple prototypes.

---

### Official Review · Reviewer_eEA9 · 2024-11-03

**Soundness:** 3
**Presentation:** 3
**Contribution:** 2
**Rating:** 5
**Confidence:** 4

**Summary:**

This paper proposes a prototype-based network for explainable recommendations. The key model pipeline involves (1) prototypes generated by a controllable music generation model (MusicGen), (2) transformed user history into an interpretable sequence, where multi-head attention was used to learn the weight distributions over prototypes, and (3) a feed-forward neural network for the recommendation task as an extreme classification problem, where two auxiliary objectives accompany the recommendation objective: a controllability objective to impose calibration of user-level tag preference, and a prototype-separability objective to avoid prototype collapse.

The experiments were conducted on MSD, a commonly used music recommendation dataset. The paper claims improvements in controllability through their proposed metric, which is calculated based on the difference of tag-wise ranking after removing certain prototypes, while maintaining competitive performance to the compared recommendation baselines.

**Strengths:**

This paper introduces an interesting perspective on combining prototype networks and the controllability of recommendations. The fact that the prototypes were interpretable (i.e. listenable music clips) is a nice feature, and the controllability is measured by the calibration of user tag preferences also provides a direct way to implement user controls.

The experiments seem to indicate the effectiveness of the added two objectives by improving the recommendation performance. The authors further analyzed the tag-wise performance drop to support the importance of these prototypes. A series of ablation studies were conducted to study the importance of model parameters.

**Weaknesses:**

The method lacks novelty: each component of the whole model is not new. The key concept of using prototypes for explainable recommendations has been explored in [1]. Different from [1], the number of prototypes is fixed in this paper and aligned with pre-defined song tags, which can limit the expressiveness of the model and may suffer from noisiness in tag data. The quality of these prototypes is delegated to a generative music model, but the experiments do not address details on how the quality may affect model training and controllability.

The baselines compared is rather limited: To show the recommendation performance is competitive, the authors shall also compare models with reported superior performance on the MSD dataset such as [2] (as reported in the RecVAE paper) to provide more references. I also recommend the authors consider comparing to ProtoMF [1], since method-wise they also used the concept of prototype network for recommendations.

[1] Alessandro B. Melchiorre, Navid Rekabsaz, Christian Ganhör, and Markus Schedl. 2022. ProtoMF: Prototype-based Matrix Factorization for Effective and Explainable Recommendations. In Proceedings of the 16th ACM Conference on Recommender Systems (RecSys '22). ACM, 246–256.
[2] Harald Steck. 2019. Embarrassingly Shallow Autoencoders for Sparse Data. In The World Wide Web Conference. ACM, 3251–3257.

**Questions:**

1. Can you explain more about the process of prototype generation, in terms of why choosing a certain generative model, the diversity (similarity between generated prototypes), and quality (tag-music alignment) of the generated prototypes? Furthermore, how it may affect the performance of the resulting recommendation model?

2. Can you further elaborate on the ablation study on training loss components (sec 5.4), in particular, was the monotonically increasing recommendation performance suggesting the recommendation model without the regularization terms was badly trained? If you add more searches on $\lambda_{1}$ and $\lambda_{2}$, will you see a tradeoff between the multiple objectives?

---

> ### Author Response · Authors · 2024-11-28
>
> We would like to thank the reviewer for their time creating their constructive feedback. Below are our responses to individual comments:
>
> Weakness 1 (Lacks novelty): Apart from the ProtoMF paper, we are using audio representations instead of learning user and item representations from the interactions initialized as free parameters. In our method, we used tag-based prototypes to express the user profiles in terms of prototypes that correspond to music tags. This enables us to control the user profiles after the model is learnt: A user would be able to listen to the prototypes that correspond to the learnt profile and remove/add prototypes to steer the system in the direction of their own preferences.
>
> Question 1 (Prototype generation): We would like to note that we used a generative model because we have the tag information in our dataset and via a generative model we are able to generate listenable prototypes that correspond to a particular tag. This way we associate the information with listenable prototypes.
>
> Weakness 2 (Baselines): It is not possible to directly compare the ProtoMF model since they are doing weak-generalization (train, valid and test users are the same.). We provide the EASE results as follows:
> | Method        | Recall@20 | Recall@50 | NDCG@100 |
> |---------------|-----------|-----------|----------|
> | EASE          | 0.333     | 0.428     | 0.389    |
> | MultiDAE      | 0.253     | 0.355     | 0.300    |
> | MultiVAE      | 0.264     | 0.366     | 0.315    |
> | RecVAE        | 0.275     | 0.373     | 0.325    |
> | MacridVAE     | 0.276     | 0.369     | 0.330    |
> | SEM-MacridVAE | 0.290     | 0.384     | 0.343    |
> | APRON (Ours)  | 0.273     | 0.371     | 0.321    |
>
> We see that even though the results obtained with EASE are better, we would like to emphasize that EASE is not controllable, and does not allow for cold-starts, and there it has to be retrained since the method outputs $Item \times Item$ matrix of coefficients.
>
> Question 2 (More Ablation Experiments): We ran additional experiments for different lambda configurations. We observed that using smaller lambda_2 values decreases controllability, on the other hand using larger lambda_2 values decreases the overall recommender system performance which also leads to lowered controllability. Lambda_1 is also important since it boosts overall recommender system performance in all settings.
> | λ_1 |   λ_2  | Recall@20 | Recall@50 | NDCG@100 |   ∆@20   |
> |:---:|:------:|:---------:|:---------:|:--------:|:--------:|
> |  0  |    0   |   0.246   |   0.335   |   0.293  |  0.00010 |
> | 0.1 |    0   |   0.245   |   0.334   |   0.292  | −0.00100 |
> |  0  |  0.005 |   0.266   |   0.362   |   0.315  |  0.03646 |
> |  1  |  0.005 |   0.273   |   0.371   |   0.321  |  0.04687 |
> |  0  |  0.05  |   0.226   |   0.308   |   0.273  |  0.02279 |
> |  0  | 0.0005 |   0.252   |   0.342   |   0.299  |  0.01259 |
> |  1  |  0.05  |   0.252   |   0.343   |    0.3   |  0.03253 |
> |  1  | 0.0005 |   0.253   |   0.345   |   0.302  |  0.01263 |

---

### Official Review · Reviewer_kZ1T · 2024-11-04

**Soundness:** 3
**Presentation:** 2
**Contribution:** 2
**Rating:** 5
**Confidence:** 5

**Summary:**

This paper proposes a music recommendation method based on audio prototypes. The method first inputs the music tags into the music generation model to generate the corresponding audio; then the MERT model is used to extract the final feature of 1024 dimensions of each generated audio. During the training process, each song in the user's behavior obtains the weighted representation of tag-audios through attention network; the user's representation is the sum of all behaviors(songs) and used for recommendation in final.

**Strengths:**

1. This paper attempts to use audio features for music recommendation to solve the problem of insufficient keywords/tags problem.
2. This paper proposes a metric to test the controllability of the model.

**Weaknesses:**

1. Although the author proposes that the model is controllable, the review is not clear about the definition of controllability in the paper, and the comparison between the model and other baseline methods (only one), and where its controllability is superior.
2. There are many classic CF methods and DL-based methods for recommendation based on user behavior. The reviewer noticed that the paper only selected VAE-based methods for comparison without any explanation or motivation of such selection.
3. The paper emphasizes the use of prototype audio of music for recommendation. These audios are generated by the original tags of the music via MusicGen, but the author does not explain the details and settings of the generation process, not provide analysis and distribution of the generated audio, and cannot provide proof of whether there is an essential difference between these audios and the original tags in the role of recommendation, nor does it provide comparative experiments to prove the difference between the two. Therefore, the review cannot determine that the paper solves the limitation problem of tag-based recommendation proposed by audio-based recommendation.

**Questions:**

The main questions are the problems listed in the weakness. In addition, there is a technical question in line140 equation 3 and line148 equation 4. Are the  $x^i_j$  in the two equations the same or not? How is the query $x^i_j$ in the attention represented?

---

> ### Author Response · Authors · 2024-11-28
>
> We would like to thank the reviewer for their time creating their constructive feedback. Below are our responses to individual comments:
>
> Weakness 1 (Controllability): Since there is no definition for controllability for this specific problem in the literature we define it in this paper (Equation 17 in the paper), where we calculate the change in NDCG when a prototype that corresponds to a particular tag is removed. We have compared it with SEM-MacridVAE as it is the most appropriate method for this comparison. We observe that when we reduce or increase the attention weights that correspond to a particular tag, the model is able to follow the change in the attention weights (Figures 3, 4, in the paper).
>
> Weakness 2 (Baseline Selection): In our strong generalization data split, where we have non-overlapping users in train, valid and test, we aimed to report competitive baselines as other baselines did so we reported these baselines. Also, our method is compatible to compare with Autoencoder-based models that is why we included them as baselines. We also would like to note that in the VAE recommendation-system papers that we cite, the authors compared with several other baselines.
>
> Weakness 3 (Musical Aspects): The purpose behind using prototypes is that we would like to be able to capture various musical nuances that can only be represented with the audio modality. (E.g. certain rhythmic, melodic, harmonic patterns. We agree with the reviewer that in the current version, since we only have one template per tag, the current system is not significantly different from tag-based prototypes. However, we believe it serves as a proof-of-concept for future improvements that use multiple prototypes.
>
> Question 1: There is a typo in Equation 3. What we wanted to express is that $x_j^i$ was approximated as a linear combination of prototypes.

---

### Official Review · Reviewer_S6hu · 2024-11-08

**Soundness:** 2
**Presentation:** 2
**Contribution:** 2
**Rating:** 3
**Confidence:** 4

**Summary:**

This paper aims to tackle the task of controllable music recommendation. It leverage music clip level prototypes as explanations and propose an attention-based recommendation model to fulfil the task. Experiments are conducted to demonstrate how this method performs the control during recommendation. Multiple experiments illustrate several properties of this method.

**Strengths:**

* The task of controllable music recommendation is valuable in both academia and industry.
* The motivation of using music-clip level prototypes is reasonable and clear. And the way to directly use music content for recommendation is a promising direction.
* The writing is clear and easy to follow.

**Weaknesses:**

1. Less technical novelty:
* The proposed prototype-based controllable music recommender model is a quite straightforward attention-based neural network model with certain losses. The attention-based model architecture has been proposed and extensively studied in recommender systems, which even though is practical and helpful,  it is not quite novel to the research or industry community.
* The learning or extraction of the prototype is based on some existing methods (MERT or MusicGen). I am expecting certain innovations in this part, which is quite interesting. For example, is it possible to automatically mine such prototypes purely based on user behaviours’ data as supervision? Furthermore, the music understanding model that is used to extract such prototypes should also be optimized during this process (in either end2end or multi-stage manners)?

2. Lacking rigorous evaluation:
* Only one dataset. I understand that it is not easy to obtain many datasets pertaining to this task formulation, while only using one dataset is less convincing to justify the generalization capability of the proposed method.
* The baselines are just general music recommendation models, which could be weak. Explainable recommendation is a quite popular topic in the past few years, and I assume there should be quite a number of works that can be implemented or adapted to this task, while they are not included in this paper. In addition to performance comparison with baselines, the controllability comparison with baselines should be also considered.
* For Section 5.5, there are many cold-start recommendation methods that can be implemented as baselines. I hope you can select some methods to do a comparison study instead of only comparing with the model itself in the non-cold start setting.
* Even though the target of this paper is the controllability rather than overall performance, it is still a concern that the quantitative performance of the model is not very strong.

**Questions:**

Please see the above comments.

**Details Of Ethics Concerns:**

There is no ethics concerns.

---

> ### Author Response · Authors · 2024-11-28
>
> We would like to thank the reviewer for their insightful feedback. Below we address the weaknesses that you point out in your review:
>
> Weakness 1 (Less technical novelty): While we agree with the reviewer that attention-based models are commonplace in the recommendation system literature, the aim of our work and our contribution is to use the attention mechanism to create a bottleneck through controllable and listenable prototypes using audio features (the novel component).
>
> Weakness 2 (Less technical novelty - Pretrained Models): In this submission, we explore using a text-prompted generative model to create these prototypes. However, it is certainly interesting to explore this research direction to mine prototypes based on user behaviors. As a future direction, fine-tuning MERT model with LoRa could be an alternative. We would like to note that this work serves as a proof-of-concept for future potential follow-up work that can incorporate the ideas along the aforementioned direction.
>
> Weakness 3 (Lacking rigorous evaluation - Another Dataset): Although some other audio recommender datasets exist, they don’t have raw audio files in which our method requires, so for now we are limited to only Million Song Dataset. We tried to match “Last.fm Dataset - 1K users” preference data(http://ocelma.net/MusicRecommendationDataset/lastfm-1K.html) with audio files, but couldn’t as the track IDs do not match.
>
> Weakness 4 (Lacking rigorous evaluation - Baselines): As you mentioned we have included baselines such as MultiDAE, MultiVAE, RecVAE, MacridVAE, and SEM-MacridVAE. Note that we have also provided a comparison with SEM-MacridVAE for controllability. The other recommendation systems baselines we considered are not compatible for the same type of controllability as they are not using prototypes.
> Also, we would like to emphasize that our model differs from post-hoc explainable methods as we are aiming to control and explain user behaviors through training, which is different from explaining model decisions.
>
> Weakness 5 (Lacking rigorous evaluation - Cold Start):  We have not compared with another cold-start baseline, because our primary purpose in doing so was to show that our method, in addition to being controllable, can also be used for cold-start. As we have discussed above (e.g. weakness 4) most of the other baselines do not offer prototype-based controllability that we seek in this paper.
>
> Weakness 6 (Lacking rigorous evaluation): As you have pointed out our priority is on controllability in this paper.

---

### Meta-Review · Area_Chair_mRqL · 2024-12-21

**Metareview:**

This paper proposes the audio prototypical network method for controllable music recommendation. The motivation to improve the interpretability and controllability of music recommendations is recognized by the reviewers, and the paper is easy to understand. However, reviewers also pointed out some key weaknesses, including: 1) Less technical novelty, as attention-based neural networks can achieve similar results; 2) Insufficient experiments, especially in terms of baseline selection and dataset (currently only one dataset is used); furthermore, there is controversy over the controllability metrics, and the final model's accuracy did not outperform existing models; 3) Unclear differences with existing settings. Since there is only one template per tag, the system is not significantly different from tag-based prototypes. As a result, all reviewers believe the paper is not ready for publication by far.

**Additional Comments On Reviewer Discussion:**

Although the authors responded to these issues during the rebuttal phase, the reviewers did not provide direct replies. After reviewing, I believe that the challenges have not been fully addressed, so I recommend rejection.

---

### Decision · Program_Chairs · 2025-01-22

Reject